# Determination of l-Asparaginase Activity and Its Therapeutic Monitoring in Children with Hematological Malignancies in a Single Croatian Center

**DOI:** 10.3390/diagnostics14060623

**Published:** 2024-03-15

**Authors:** Jasna Lenicek Krleza, Ana Katusic Bojanac, Gordana Jakovljevic

**Affiliations:** 1Department of Laboratory Diagnostics, Children’s Hospital Zagreb, 10000 Zagreb, Croatia; 2Department of Basic Medical Chemistry and Biochemistry, University Studies in Nursing, Universitas Studiorum Catholica Croatica, 10000 Zagreb, Croatia; 3Department of Medical Biology, School of Medicine, University of Zagreb, 10000 Zagreb, Croatia; 4Centre of Excellence for Reproductive and Regenerative Medicine, School of Medicine, University of Zagreb, 10000 Zagreb, Croatia; 5Department of Oncology and Hematology, Children’s Hospital Zagreb, 10000 Zagreb, Croatia; 6Department of Pediatrics, Faculty of Medicine, University of Osijek, 31000 Osijek, Croatia

**Keywords:** asparaginase, asparaginase enzyme activity, therapeutic drug monitoring, lymphoblastic malignancy, child

## Abstract

Background: Among malignant diseases which develop during childhood, hematological cancers, such as leukemias and lymphomas, are the most common. Outcomes have greatly improved due to the refinement of multiagent chemotherapy regimens that include enhanced asparaginase therapy. In this study, we aimed to evaluate our experiences related to the analytical and clinical significance of determining l-Asparaginase activity. Methods: Since 2016, the Laboratory of the Children’s Hospital Zagreb has routinely measured l-Asparaginase activity and to date, has measured more than 280 examples of activity in a total of 57 children with hematological malignancy treated at the Pediatric Oncology Department of the Children’s Hospital Zagreb. Three asparaginase products were available: native *E. coli*
l-Asparaginase; a pegylated form of this enzyme; and a native product from Erwinia chrysanthemi. A retrospective data analysis was performed. Results: Out of the fifty-seven children, seven had an allergic reaction (12.3%), five (8.8%) had silent inactivation, and seven (12.3%) developed acute pancreatitis. Allergic reactions and silent inactivation were more common in children treated with native *E. coli*
l-Asparaginase, while pancreatitis was more common in children treated with the pegylated form. Conclusions: The monitoring of l-Asparaginase activity may help to optimize therapy by identifying patients with ‘silent inactivation’, and/or by dose correction when l-Asparaginase activity is too high (slow elimination).

## 1. Introduction

Among the malignant diseases that develop in childhood, hematologic cancers such as leukemias and lymphomas are the most common. A total of 75% of all childhood leukemias are classified as acute lymphoblastic leukemia (ALL), while among lymphomas, non-Hodgkin’s lymphoma (NHL) is the most frequently observed [1].

In the time period of this study, the available published data from the Croatian Cancer Registry (2015–2021) recorded an average of 26 new ALL cases in children (0–19 years old) per year [1,2], while in the previous eight years (2010–2018) 147 new cases of ALL in children were reported, which amounts to an average of 18 new cases of ALL in children per year [3]. The data on the incidence of new cases of ALL and NHL in Croatian children are shown in Table 1 [1,2,3].

According to these latest data, 15–35 new cases of ALL per year and 2–7 new cases of NHL per year are reported in Croatian children (0–19 years old). The extensive global observational study CONCORD-3 shows that five-year survival from malignant diseases in children in Croatia is above the European average. For childhood lymphoma, the average survival rate is 95% and for ALL, 85%, with the latter varying depending on the risk group [4].

Today’s high survival rate of >90%, compared to the 1960s when this rate was <30%, is mostly due to better chemotherapy protocols which use multiple agents. For decades, l-Asparaginase (l-asp) has been proposed and used as one of the agents against hematological cancers [5,6,7,8].

The mechanism of action of l-asp is based on l-asp enzyme activity that catalyzes the deamination of asparagine into aspartic acid and ammonia, thereby reducing the concentration of asparagine in the blood. As lymphoblastic leukemia cells do not have the ability to synthesize asparagine, they depend on exogenous asparagine for protein synthesis and cell growth. With a constant, sufficient level of l-asp in the blood, the activity of the enzyme will lead to the complete depletion of the concentration of asparagine, which results in reduced protein synthesis and, finally, the death of the malignant lymphocytes [6,9,10].

Three types of l-asps are used for treatment of ALL: two types derived from *E. coli* (native l-asp) and pegylated l-asp (PEG l-asp), as well as one derived from Erwinia chrysanthemi [10].

Native *E. coli*
l-asp (Kidrolase^®^) is widely used as the first line treatment in ALL. However, the supply of this preparation was recently stopped (in 2020) in Croatia and has mainly been replaced by PEG l-asp (Oncaspar^®^). The bacterial origin of Erwinia asparaginase (Erwinase^®^) gives it a unique immunogenic profile: it has no cross-reactivity with native *E. coli*
l-asp or PEG l-asp [11]. Because of this property, Erwinia asparaginase is indicated as a component of a multiagent chemotherapy regimen for the treatment patients with ALL who have developed potential hypersensitivity to l-asp derived from *E. coli* [6,12].

Each of the three l-asp preparations (native or pegylated l-Asparaginasec, and Erwinia l-asp) have the same mechanism of action, efficacy and side effects, but show significantly different pharmacokinetics that must be considered when determining the dosage schedule. The half-life of PEG l-asp is estimated to be 4.8–7.0 days, while that of native *E. coli*
l-asp and Erwinia l-asp is 1.28 days and 15.6 h, respectively. Due to the shorter half-life of Erwinia l-asp, patients switching to this formulation should receive a higher dose at a higher frequency to maintain therapeutic levels of asparagine deficiency [13,14].

Hypersensitivity is mostly based on the secretion of antibodies against l-asp and can be manifested to clinically observed allergic reactions (AR). It sometimes occurs without any clinically apparent allergic reaction, which is described as a phenomenon of silent inactivation (SI). The production of anti-asparaginase antibodies has been investigated by many authors, who have reported that various IgG molecules were present in serum samples of patients with asparaginase allergies [15,16,17,18].

However, intensive use of l-asp is associated with numerous toxic effects, and in cases where the full course of therapy with l-asp is not carried out due to treatment of this toxicity, it is associated with poor outcomes in children with ALL [6,7,8,9]. Other than immune reactions to l-asp, the use of l-asp is also associated with toxicities such as pancreatitis, thrombosis, hyperglycemia, hepatic toxicity, hypertriglyceridemia, etc. [6].

Since November 2016, the Department of Laboratory Diagnostics of the Children’s Hospital Zagreb has introduced asparaginase monitoring in the plasma of individuals treated with asparaginase, thereby becoming the first laboratory in Croatia to introduce the determination of l-asp activity into routine work.

In this study, we aimed to evaluate our experiences related to the analytical and clinical significance of determining l-asp activity in patients with ALL and NHL treated at the Department of Pediatric Oncology of the Children’s Hospital Zagreb, along with the summarization of the occurrence of AR, SI or other toxicities related to several asparaginase preparations from different sources.

## 2. Materials and Methods

### 2.1. Study Population

In the period from September 2015 to January 2024, all children up to 18 years of age diagnosed with ALL (*N* = 53) and NHL (*N* = 4) who were being treated at the Department of Pediatric Oncology of the Children’s Hospital Zagreb, were included in this prospective l-asp study. The measurement of l-asp activity was carried out at the Department of Laboratory Diagnostics of the Children’s Hospital Zagreb, Croatia (CHZ_LAB). The total number of determined examples of l-asp activity at anywhere between one and seven time points of therapeutic drug monitoring was 237.

### 2.2. Asparaginase Treatment

In the study period from September 2015 to December 2020 (5 years), the first line of treatment for hematological malignancies was native *E. coli*
l-asp (Kidrolase; *N* = 33), but three patients started treatment with pegylated native *E. coli* (Oncaspar: *N* = 33), and from April 2021 to January 2024 (3 years), all the patients received pegylated native *E. coli* in the first line of treatment (Oncaspar; *N* = 21). Erwinia l-asp (Erwinase; *N* = 11) was used only as a second-line drug in cases of allergic reaction or silent inactivation during the eight-year study period.

In our study, the ALL-IC-BFM-2009 protocol was used for patients with de novo ALL. These patients were stratified into three risk groups: standard risk (SR), medium risk (MR), and high risk (HR). For children with relapsed ALL, the ALL-IC-REC-2016 protocol was used, and for lymphoblastic lymphoma patients the LBL-2009 protocol was used. According to the ALL-IC-BFM-2009 protocol, Kidrolase was used during the induction phase (5000 IU/m^2^/dose/2 h infusion), every third day, 8 times, starting on day 12. In the reinduction phase, Kidrolase 10,000 IU/m^2^/dose/2 h infusion × 4 times was used. Since April 2021, instead of Kidrolase, 1500 UI/m^2^/dose Oncaspar was administered in 2 h infusions, twice in 14-day intervals during the induction phase, and once in reinduction (on day 7). The HR patients received 6 times 1500 UI/m^2^/dose of Oncaspar for HR blocks. The relapsed ALL patients received no Kidrolase, but Oncaspar once in each block. In the patients who developed clinical hypersensitivity or silent inactivation (SI), asparaginase therapy switched to an alternate asparaginase preparation. In the case of a Kidrolase hypersensitivity reaction or SI, Oncaspar was administered (1500 UI/m^2^/dose replaced 4 Kidrolase). In the case of an allergic reaction or SI for Oncaspar, Erwinase was administered in a dose of 10,000 IU/m^2^/dose every second day. One dose of Oncaspar was replaced with seven doses of Erwinase. From 2021, we corrected Oncaspar doses according to recommendations based on therapeutic drug monitoring (TDM) [19].

The study also included three patients treated with native *E. coli*
l-Asparaginase (Kidrolase) who were randomized to early intensification augmented asparaginase from the start of the study, whereas this form of early intensification was not used in patients treated with pegylated *E. coli* asparaginase (Oncaspar).

### 2.3. Asparaginase Enzyme Activity Measurement

Blood samples were collected at the beginning and during treatment according to protocol (Table 2). In the process of introducing and verifying the method in CHZ_LAB, l-asp activity in the serum and plasma samples was tested. After verification, the sample of choice was serum.

For serum, 3.5 mL of blood was collected in a tube without anticoagulant (VACUETTE^®^ TUBE CAT Serum Separator Clot Activator, gold cap), and for plasma, 2 mL of blood was collected in a tube with K3EDTA anticoagulant (VACUETTE^®^ TUBE, lavender cap, by Greiner Bio-One, Kremsmünster, Austria). Serum and/or plasma were separated after centrifugation for 10 min on 1300× *g* and were stored at −20 °C until analysis (for short-term storage up to one month at −20 °C, or for long-term storage, which means more than one month and a maximum six months in the period of method verification, at −70 °C).

For measurements of asparaginase activity, we used a modified indooxine method developed by Lanvers et al. [20].

In brief, a 20 µL of sample or standard was diluted with 180 µL 10 mM L-aspartic acid ß-hydroxamate (AHA) solution in 15 mM Tris buffer at pH 7.3, supplemented with 0.015% (*w*/*v*) of bovine serum albumin (BSA). The samples were incubated for 10 min at 37 °C, and the reaction was later stopped with the addition of 60 µL of 24.5% (*w*/*v*) TCA. The mixture was then centrifuged at 2500× *g*, after which 20 µL of the supernatant was removed and diluted in a 1:3 volume ratio with deionized water. To reveal the absorbance, the samples were incubated with 200 µL of an 8-hydroxyquinoline solution (1 mL of 2% of 8-hydroxyquinoline in 3 mL 1 M Na2CO3) at 95 °C for 1 min. After cooling, the absorbance was read at λ = 690 nm on a Tunable Microplater Reader VersaMax with SoftMax Pro software (ver 6.2.1. 2015, Molecular Devices, LLC., San Jose, CA 95134, USA). The reading time was optimized after 10 min [21].

### 2.4. Statistical and Analytical Methods

The result comparability was tested via the paired samples t-test using MedCalc^®^ statistical software (MedCalc Software Ltd., version 22.017, Ostend, Belgium; https://www.medcalc.org; (updated and accessed on 6 March 2024), which was also used to test the differences between the study groups using a *t*-test. Thus, the mean and median *t*-test was performed to test the difference between age groups compared to diagnosis (Table 3), as well as to test the significance of the difference between the mean values of l-asp activity in the TDM points (Table 4). The comparison test of proportions was used to test the significance of the difference in the occurrence of toxic effects of l-asp therapy depending on the type of preparation (Table 3). Summary statistics (descriptive data) were calculated using the same MedCalc software program and are presented in Table 4 and Table 5. Graphs (Figure 1 and Figure 2) were also created using the MedCalc program (multiple variable graph, box-and-whisker plot with all data). The normality of the distribution of the individual variables was tested as part of the summary statistics using the Kolmogorov–Smirnov test.

Microsoft Excel 2016 was used to keep the obtained results and records of all data of the patients, as well as their calculation, analyses and error-bar graph (the bar graph with the standard deviation marker in Figure 3).

The statistical significance of the difference was defined at the level of probability *p* < 0.05.

Direct measurement of anti-L-asp antibodies was not available in Croatia during this study. Low l-asp activity (<100 IU/L) without a clinically visible allergic reaction is defined as silent inactivation (SI).

## 3. Results

### 3.1. Analytical Characteristics of Method

The procedure for introducing the method defined that the sample for the measurement of l-asp activity can be either serum or plasma. In our conditions, the first choice of sample was serum. Furthermore, we confirmed that the activity of l-asp will not change if the sample is analyzed within 2 h of sampling or stored short-term (up to 1 month) at −20 °C, or long-term (>1 month, up to 6 months) at −70 °C. For sample transport longer than 2 h, it was necessary to keep the sample frozen using dry ice. The optimal time to read results was 10 min after the addition of Oxin reagent (at a working room temperature of approximately 24 °C). Standards and control samples were prepared from a commercial drug; their stability is 6 months if aliquots are stored at −70 °C. The interlaboratory comparison of the obtained results showed excellent comparability (*p* = 0.5801) (Figure 1).

### 3.2. Patient Characteristics

A total of 237 examples of l-asp activity was measured at 1–7 points during treatment in a total of 57 patients.

In the eight-year period of our study, 93% of children with hematological malignancy had ALL de novo and 7% of children had NHL. In this study, there were slightly more girls (*N* = 30) than boys (*N* = 27). The average age of all the children included in the study was 5 years 9 months (with a range from a minimum of 2 months, to 16 years and 4 months). Although our results do not show statistically significant differences in the mean age of children with ALL and NHL (PAM = 0.0716), older children had a higher prevalence of NHL, while those of younger age had ALL (when the median was compared, PMED = 0.0145). There was no significant difference in the incidence of AR (5/36) and SI (4/36) during the period when Kidrolase was the first line of treatment (5 y), and in the following period (3 y) when treatment was started with Oncaspar (*p* = 0.5448). However, the incidence of pancreatitis (PA) was significantly higher in patients treated with Oncaspar (*p* = 0.0455). Table 3 presents a detailed overview of the patients’ characteristics during treatment with l-Asparaginase, including the incidence of the main toxic effects (allergic reaction, silent inactivation, and pancreatitis).

Table 4 shows in detail all important statistical indicators of l-asp activity (average, median, standard deviation, range of minimum–maximum activity, and difference) in different time points of the monitoring and according to the type of l-asp preparation. Figure 2 shows all the data with the trend (according to the average l-asp activity) of decreasing l-asp activity at follow-up time points. Figure 3 shows PEG l-asp activity at follow-up time points in five patients who developed pancreatitis during l-asp treatment, with tabular presentation of statistical data in Table 5.

## 4. Discussion

The aim of this study is to present our experience with the measurement of l-asp activity over the past eight years during the treatment of children with ALL and NHL, with several new key moments: (1) the successful introduction of the measurement of l-asp activity into the routine work of our laboratory; (2) therapeutic drug monitoring (TDM) activity of l-asp during treatment according to regional protocols; (3) changes in the drug of choice in the first line (native *E. coli*
l-asp was replaced by PEG l-asp); (4) dose correction in the treatment protocol with PEG l-asp, and (5) defining the value of l-asp activity for drug ineffectiveness and substitutes for Erwinase.

The epidemiology of leukemia may change over time, just as possible variability in incidence, as well as gender, age, global, and regional differences from country to country may occur. Therefore, these data are important for the planning and development of treatment, prevention, and survival policies. Hence, the incidence and demographic data of our patients are an integral part of the study [2,22].

According to our data (Table 1 and Table 3), and the data of the Croatian Cancer Registry at the Department of Pediatric Oncology of the Children’s Hospital Zagreb, an average of seven new cases per year (under the age of 18) were treated, which is about quarter of the total number in Croatia [1,2].

During the study, an equal number of girls and boys were treated in our center. However, there were slightly more girls (as shown in Table 3: boys/girls—27/30). Gender differences related to the risk of childhood ALL, described in the literature, globally show a higher incidence rate in boys compared to girls, although the reason for this is still unknown [2,23,24,25].

Hematological cancers can be diagnosed at any age, but ALL is the most common malignancy in children and in our center, 93% of respondents with a hematological malignancy were diagnosed with ALL, while 7% had NHL. These frequency data are also in accordance with published results at the national level of a neighboring country in the region. Also, although they occur in all age groups, the median age of children with ALL (4 y 8 m) is significantly lower than the median age of children with NHL (10 y 6 m) (Table 2; *p* = 0.0145) [23,25].

The focus of our research was twofold: analytical and clinical. This necessitated the successful verification of the method of determining the activity of l-asp and its introduction into the routine work of the laboratory, as well as successful therapeutic drug monitoring in real time according to the protocol of treatment with different preparations of l-asp.

Our results showed that during the first 5 years of monitoring the activity of l-asp, native *E. coli*
l-asp was the first drug of choice; in that period, according to the treatment protocol, the mean activity of l-asp was 241.4 U/L (48 h after the dose), and 314.1 U/L (72 h after dose). During this five-year period using native *E. coli*
l-asp (Kidrolase) we recorded a greater number of allergic reactions and silent inactivation and, as a consequence, we switched to Erwinase. However, in the next three-year period (2021–2024), the first drug of choice was pegylated native l-asp (Oncaspar), which resulted in a lower incidence of allergies and silent inactivation, but an increase in pancreatitis.

Asparaginase is a protein of high molecular weight and is not excreted by the kidneys, while proteolytic enzymes in the tissues are responsible for l-asp metabolism. Inactivation of *E. coli*
l-asp can be detected in up to 60% of cases due to anti-asparaginase antibodies [16]. These antibodies rapidly neutralize circulating l-asp and cause clinical hypersensitivity (allergic) reactions, but sometimes this occurs without any clinically apparent allergic reaction (AR), a response that is called silent inactivation (SI). SI prevalence in different studies is in the range from 8% to 30% [15,16], with the consequence being the lowered concentration and activity of drugs in plasma or serum. The monitoring of l-asp activity in a patient’s blood sample is therefore necessary to find patients with inadequate l-asp activity and, consequently, decreased clinical efficacy of prescribed drugs. l-asp activity above 100 IU/L was shown to lead to the complete depletion of asparagine in blood. Later, many clinical trials confirmed this cut-off value of >100 U/L for essential clinical efficacy of l-asp [16,17,18].

However, in this period, more pancreatitis was reported as a significant side effect.

Two cases of pancreatitis in the period of the first five years were recorded with high l-asp activity: 795.6 U/L 48 h after the use of Kidrolase (the average activity of l-asp when using Kidrolase is 326 U/L), and another case after high l-asp activity of 152.7 U/L 48 h after intramuscular administration of Erwinase (the average l-asp activity after administration of Erwinase is 95.5 U/L).

There were five cases of pancreatitis in the three-year period when PEG l-asp was applied. The first three cases during 2021, after 7 days of drug administration, had l-asp activity > 1000 U/L (the average in these three cases was 1241.2 U/L, as seen in Figure 3), which is significantly higher activity compared to average l-asp activity after 7 days of dosing (747.9 U/L) (*p* = 0.0078). At the end of 2021, a dose correction was introduced according to the protocol described by Kloos R.Q.H. et al. [19]. We recorded two cases after the introduction of therapy individualization, i.e., dose correction based on therapy monitoring. Therefore, the average PEG l-asp activity of all five cases did not show a significant difference in l-asp activity (*p* = 0.1043).

Furthermore, the occurrence of allergy, silent inactivation, or pancreatitis is not necessarily a side effect of l-asp, given that l-asp is only part of a complex, multiagent chemotherapy, so to prove the connection between high values of l-asp activity and the toxic side effects of the drug, an extensive study and/or multicenter prospective systematic work are necessary.

The percentage of AR in our study (7/57; 12.3%) and SI (5/57; 8.8%) is in a wide range of percentages reported by other authors. The percentage of AR published for native *E. coli*
l-asp is 3–37%, with a slightly lower range for PEG l-asp (3–24%) [26,27]. The results from a neighboring country have reported 8.5% AR and 12% SI in contrast to our results, which are just the reverse [23]. Pancreatitis in our study was reported in 7/57 (12.3%) patients which is slightly higher than the available published figure of 7% [26,28].

Our research also has some limitations related to the relatively small number of subjects (approximately a quarter of children in Croatia with ALL are treated at our Department of Pediatric Oncology of the Children’s Hospital Zagreb). In the first five years, the lab determined l-asp activity for patients from other hospital centers; however, these patients and their l-asp activity were not included in our study. Furthermore, during 2021, another hospital center introduced the determination of l-asp activity using another method. Although the comparability of the results of the two methods was examined and confirmed, the review and analysis of the results in this study are limited only to the patients treated at the Children’s Hospital Zagreb and their l-asp activity as determined at CHZ_LAB.

However, it is important to emphasize that some comparisons (referring to the first 5 years of the study and Kidrolase in the first line of treatment, compared to the new pegylated preparation, Oncaspar, in the continuation of the study) will not, by extending the duration of the study, change the limitation of the small number of patients, because the form of l-asp is not produced by native E.coli (Kidrolase) and these cases are final and unchangeable. At the same time, this is part of the value of these results and that is why they are presented as an equal part of our experiences in monitoring children treated with l-Asparaginase.

Also, the overall small number of subjects and, consequently, the small number of reported toxicities cannot provide reliable conclusions about the association of l-asp activity and the occurrence of hypersensitivity reactions and/or pancreatitis.

## 5. Conclusions

Our eight-year laboratory experience of measuring l-asp activity in real time has proven to be immensely useful and helpful in treating hematological cancers (ALL, NHL). The current medicine of treatment using multiagent chemotherapy with constant high l-asp activity has resulted in a high cure rate. The sustainability of this treatment with potential side effects is only possible with proper therapeutic drug monitoring (TDM). In addition to low l-asp activity that indicates the ineffectiveness of treatment (the so-called silent inactivation, which was the reason for the initial importance of TDM), today the importance of TDM can also be found in the individualization of therapy through dose optimization/correction.

## Figures and Tables

**Figure 1 diagnostics-14-00623-f001:**
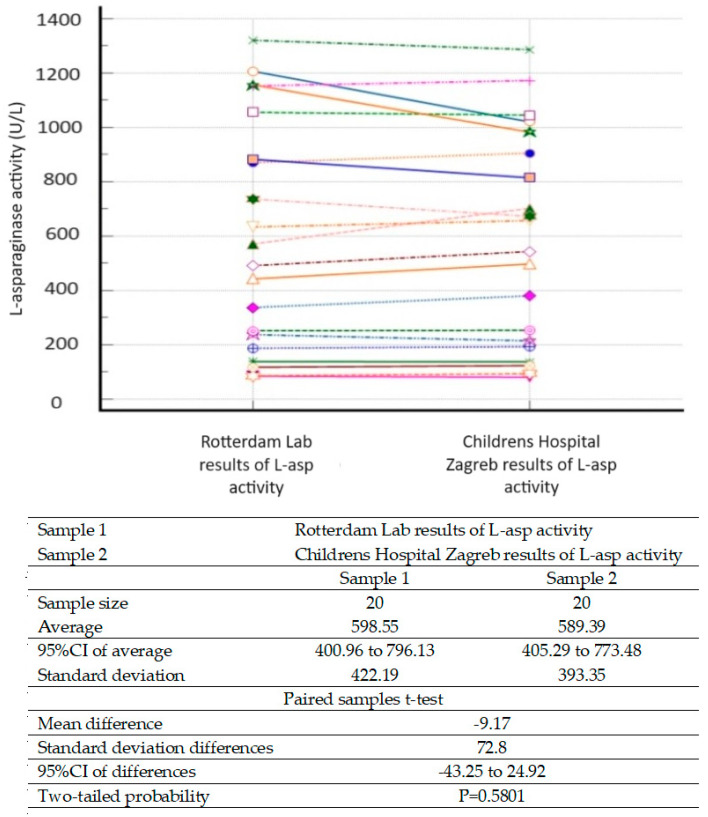
Interlaboratory comparability: the results of l-asp activity (the same samples, *N* = 20), measured in the two laboratories, the Children’s Hospital Zagreb Laboratory and the Pediatric Oncology Laboratory (Sophia Children’s Hospital, Erasmus Medical Center, Rotterdam, Netherlands), were tested by the paired samples t-test using MedCalc^®^ statistical software. The results are comparable (*p* = 0.5801).

**Figure 2 diagnostics-14-00623-f002:**
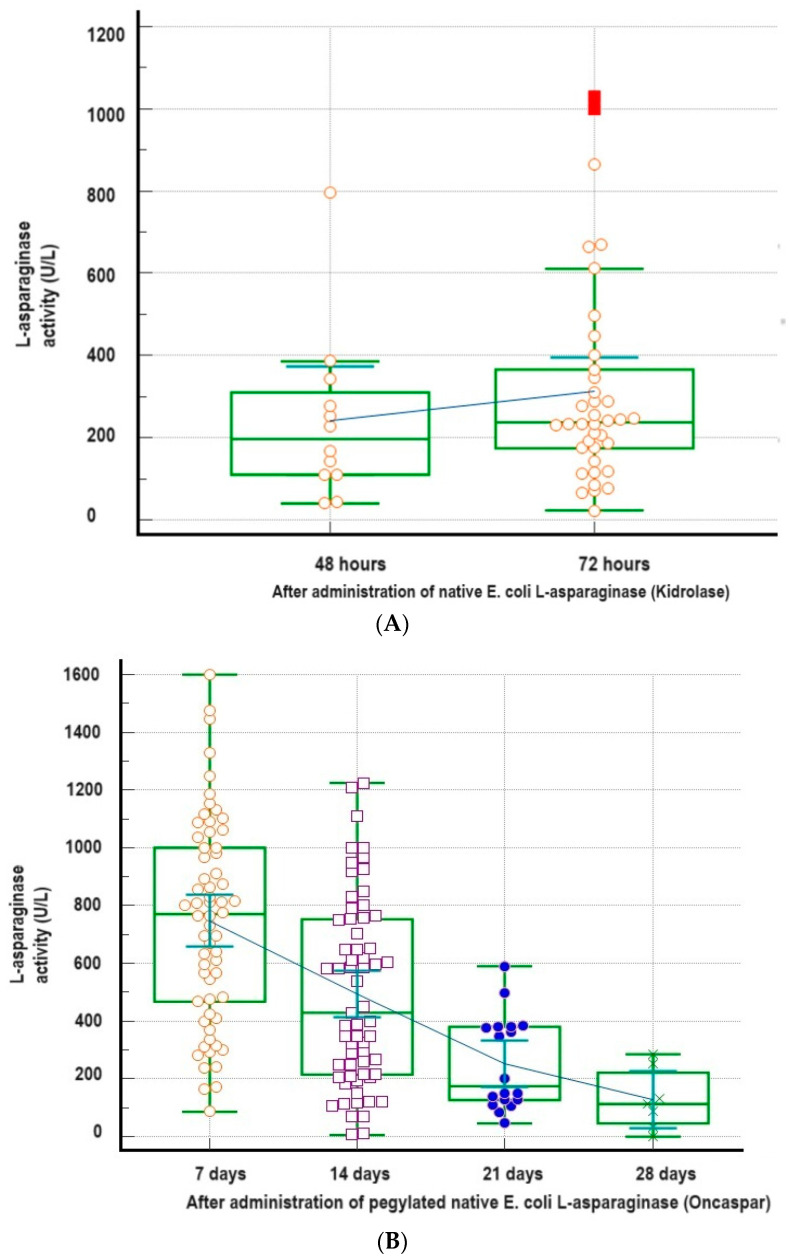
Average and statistical data of l-asp activity according to the type of l-asp: (**A**) native *E. coli*
l-asp (Kidrolase); (**B**) pegylated native *E. coli*
l-asp (Oncaspar), and (**C**) Erwinase, at a monitoring time point, as well as the trend line of decreasing l-asp activity through the follow-up time points.

**Figure 3 diagnostics-14-00623-f003:**
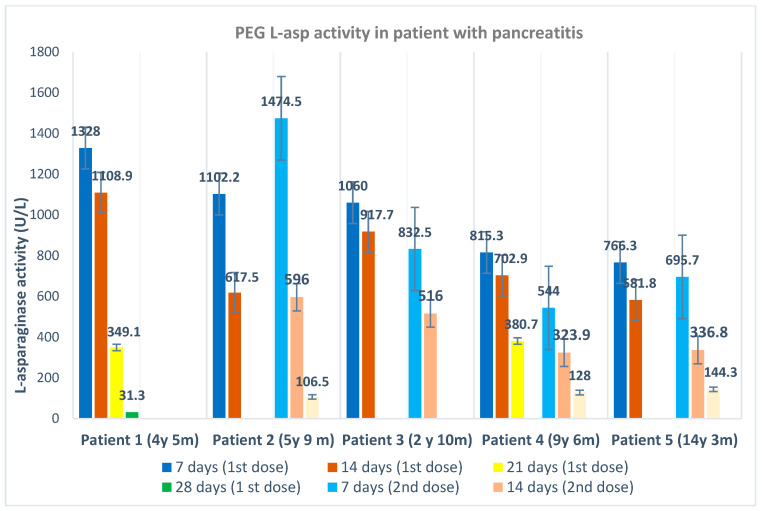
PEG l-asp activity at follow-up time points in 5 patients who developed pancreatitis during l-asp treatment.

**Table 1 diagnostics-14-00623-t001:** The data on the incidence of new cases of ALL and NHL in Croatian children with the data source.

Year	Hematology Cancers (ALL and NHL): New Cases per Year in Croatian Children (0–19 Years Old)		Source of Published Data
	ALL	NHL	
2015	35	5	Time period of this study— Croatian National Cancer Registry: Cancer incidence in Croatia. Croatian Institute of Public Health Bilten No. 40–45, Zagreb 2015–2022. Available at: “https://www.hzjz.hr/sluzba-epidemiologija-prevencija-nezaraznih-bolesti/publikacije-odjel-za-maligne-bolesti/ (accessed on 9 March 2024)” [1,2]
2016	15	4	
2017	30	7	
2018	17	5	
2019	21	5	
2020	31	5	
2021	31	2	
2022–2024	NA	NA	
Total (2015–2021)	180	33	
Average of new cases per year	26	5	
Median per year	30	5	
Range: min–max	15–35	2–7	
2010–2018	147	NA	
Average of new cases per year	147/8 (18)	NA	Campbell M et al. J Clin Oncol. 2023; 41:3499–3511 [3]

**Table 2 diagnostics-14-00623-t002:** Protocol for blood sampling.

Type of l-Asparaginase	Time of Blood Sampling
Native l-Asparaginase (Kidrolase)	48 or 72 h after dose
Pegylated l-Asparaginase (Oncaspar)	7 days and 14 days after dose (some patients have l-asp activity determined on day 21 (*N* = 18) or day (*N* = 8) after dose) *
Erwinia l-Asparaginase (Erwinase)	48 h after dose

* In patients who had high activity (l-asp > 500 U/L) 7 and 14 days after the dose.

**Table 3 diagnostics-14-00623-t003:** Characteristics of patients and incidence of l-asp toxic effects during treatment.

	September 2015 to December 2020 (5 Years)		April 2021 to January 2024 (3 Years)	
Total number of patients treated at the Department of Pediatric Oncology of the Children’s Hospital Zagreb, (*N*)	57			
Patients (*N*)	36		21	
Age for all patients				
Average (AVE) (month)	5 y 9 m (69 m)			
Median (MED) (month)	5 y 2 m (62 m)			
Range minimum–maximum (month)	2 m–16 y 4 m (196 m)			
ALL patients	AVE = 5 y 6 m (65.7 m) MED = 4 y 8 m (56 m)			P_AVE_ = 0.0716 P_MED_ = 0.0145
NHL patients	AVE = 8 y 10 m (105.5 m) MED = 10 y 6 m (120.5 m)			
Gender (number of boys/girls)	27:30			
Diagnose				
ALL (*N*)	33	20		
NHL (*N*)	3	1		
Diagnosed cancers type related to total number of study patients *N* (%)				
ALL	53/57 (93%)			
NHL	4/57 (7%)			
First line of treatment for hematological malignancies				
Native *E. coli* l-asp (Kidrolase)	33	0		
Pegylated native *E. coli* (Oncaspar)	3	21		
Switching to Erwinase	9/36 (25%)		2/21 (9.5%)	
Allergic reaction (AR) (*N*)	5/36 (13.9%)	2/21 (9.5%)		
Total AR per total *N* of patients	7/57 (12.3%)			
Silent inactivation (SI) (*N*)	4/36 (11.1%)	1/21 (4.8%)		
Total SI per total *N* of patients	5/57 (8.8%)			
Diff. of total AR and total SI in total *N*	*p* = 0.5448			
Pancreatitis (PA) (*N*)	2/36 (5.6%)		5/21 (23.8%)	
The percentage of pancreatitis related to the number of patients treated with same type of l-asp (%)				
Native *E. coli* l-asp (Kidrolase)	2/33 (6.1%)	0 (0.0%)		
Pegylated native *E. coli* (Oncaspar)	0/3 (0.0%)	5/21 (23.8%)		
Difference in PA incidence in patients treated with Kidrolase (%) and Oncaspar (%)	*p* = 0.0455			
Percentage of PA per total *N* of patients	7/57 (12.3%)			

Legend: AVE—Average; MED—Median; y—year; m—months; *N*—number; *E. coli*—Escherichia coli; ALL—acute lymphocytic leukemia; NHL—non-Hodgkin’s lymphoma; l-asp—l-Asparaginase; AR—allergic reaction; SI—silent inactivation; PA—pancreatitis; Diff.—difference.

**Table 4 diagnostics-14-00623-t004:** Statistical data of l-Asparaginase activity.

N	Time after Adminis- tration l-asp	l-Asparaginase Activity (U/L)				Difference in l-asp Activity (Mean) Compared to Previous Time Point	
		Average	Median	Range (min–max)	Standard Deviation		
						Percentage (%)	Probability (*t*-Test) (p)
Kidrolase							
12	48 h	241.4	197.1	40.2–795.6	206.4	30.3	0.3641
39	72 h	314.1	237.8	23.3–1028	248.5		
Oncaspar							
62	7 days	747.9	771.1	86.4–1599.7	353.8	33.7	0.0001
63	14 days	495.7	429.4	5.8–1224.5	320.1		
18	21 days	253.5	174.7	45.8–590.2	161.3	48.9	0.0028
7	28 days	128.9	113.1	0–285.4	106.0	49.2	0.0730
Erwinase							
36	48 h	95.5	87.4	14.2–368.1	71.1	-	-

**Table 5 diagnostics-14-00623-t005:** Statistical data of PEG l-asp activity at follow-up time points in 5 patients who developed pancreatitis during l-asp treatment.

	l-Asparaginase Activity (U/L)							
	7 Days (1st Dose)	14 Days (1st Dose)	21 Days (1st Dose)	28 Days (1st Dose)	7 Days (2nd Dose)	14 Days (2nd Dose)	21 Days (2nd Dose)	28 Days (2nd Dose)
Average	1014.4	785.8	364.9	31.3	886.7	443.2	126.3	-
Media	1060.0	702.9	364.9	31.3	764.1	426.4	128.0	-
Range (min–max)	766.3–1328.0	581.8–1108.9	349.1–380.7	31.3	544.0–1474.5	596.0–323.9	106.5–144.3	-
Standard Deviation	228.8	222.8	22.3	-	409.2	134.4	19.0	-

## Data Availability

Original data can be obtained from Jasna Lenicek Krleza upon request.

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
