# Peer review of "Determination of l-Asparaginase Activity and Its Therapeutic Monitoring in Children with Hematological Malignancies in a Single Croatian Center"

_diagnostics, 2024, doi:10.3390/diagnostics14060623_

Round 1

Reviewer 1 Report

Comments and Suggestions for Authors

General remarks.

Lenicek Krleza et al. report of the results of L-asparaginase activity monitoring of children with acute lymphoblastic leukemia (ALL) in the context of L-asparaginase-related side effects and silent inactivation from a single institution. The findings of the study are not novel. Authors state themselves that their observations are similar to many other previous studies (lines 285-287). However, this is the first study reporting on L-asparaginase activity measurements as related to asparaginase toxicity within the frames of the recently published BFM-ALL-IC-2009 (ALLIC-2009) investigator-initiated international multi-center clinical trial (Campbell M et al. J Clin Oncol. 2023; 41:3499-3511). ALLIC-2009 investigated, among else, the effect of augmented L-asparaginase treatment among pts. w intermediate and high-risk ALL in a randomized fashion, therefore present results of the authors can be considered relevant.

Detailed remarks:

No of pts given in Abstract (57; line 20) is different from that given in Materials and Methods (56; line 95) - correct.

Annual No of pts w de novo ALL in Croatia is overestimated, unless authors take adult cases into account. In the ALLIC-2009 study, Croatia has enrolled 147 pts within a period of 8 yrs, resulting in an average no of pts with de novo childhood ALL of 18 (Campbell M et al. J Clin Oncol. 2023; 41:3499-3511).

Antibodies secreted in vivo against L-asparaginase are certainly not monoclonal antibodies (line75). Not even the cited literature (item No 14) says so. Production of anti-asparaginase antibodies were investigated by many authors, including Avramis VI et al. Anticancer Res. 2009; 29:299-302, describing that various IgG molecules were present in serum samples of patients with asparaginase allergies.

Materials and Methods: When describing the study population Lines 94-99), authors did not tell if patients enrolled in the study were consecutive patients admitted to the institution because of ALL and NHL or there was certain selection of patients which might have biased the results of the study. Based on the sentences written down in the last paragraph of Discussion (lines 298-293), one can assume that the study was not comprehensive. Explain in the Materials and Methods section in more detail.

Very importantly, I did not find any mentioning of an ethical review board permission of the study, nor informed consent to be signed by pts and their caretakers!

When describing asparaginase treatment, authors wrote that they were treating patients w de novo ALL according to ALLIC 2009 clinical trial. Yet, according to their description we cannot identify patients assigned to the augmented asparaginase arms of the study (Campbell M et al. J Clin Oncol. 2023; 41:3499-3511). Were pts treated by using augmented asparaginase early intensification excluded from the presented investigations? If yes, authors should explain and include it among the limitations of their study.

Results: The ratio of AR in the first 5-yrs’ period seems to be higher than in the second 3-yrs’ period (as one could expect). However, the difference was not significant. My question is: was the study powered enough to detect a true – if any – difference between the immunogenicity of native vs. pegylated E. coli Asp-ase Lines 190-192)? If the number of patients was too small to detect such a difference, it should be mentioned among the limitation of the study.

Which statistical test was performed to check for significant differences – or the lack of it – when investigating the frequency of allergic reactions (lines 190-192) and acute pancreatitis (lines 192-194) among the two subgroups of pts? The statistical analysis performed has not been but should be described in the “Materials and Methods” section.

Conclusions: Telling that actually measuring asparaginase activity of pediatric patients treated with asparaginase is the “only correct way” of treatment (p. 298). Seems to be rather exaggerated. Many big pediatric ALL treatment protocols of excellent study groups do not perform regular asparaginase activity measurements involving all patients.

Author Response

Dear Reviewer 1, please find detailed responses to your questions/suggestions in the attached document under the tab "Answers to Reviewer 1". Thank you. Regards, Jasna Lenicek Krleza (Corresponding Author)

Reviewer 2 Report

Comments and Suggestions for Authors

The authors have conducted a retrospective study assessing the effectiveness of L-asparginase serum values for the purpose of monitoring the therapeutic response in children with blood cancer. Their results show that L-asp levels can be measured and it corelates with the silent inactivation of the enzyme or adverse events like pancreatitis. Also, with respect to the different types of L-asp administrated to the patients, they observed that the rates of allergic reactions, silent inactivation or complications variate significantly.   This might be of use for the clinicians. However, there are some downfalls of the study: the small group and the fact that there is no association with the treatment outcome of these patients.

Please address the following issues:

Line 44 - “is the average survival rate lifts to 95%”: cut out the „is”

Line 91 – What does “AI” stand for?

Line 95 there are 56 patients included in the study while at line 181 there 57. When counting No. Kidrolase + Oncaspar + Erwinase = 69. It is confusing. Please clarify

 Line 268 and 273: Is the association between pancreatitis and the high levels of L-asp in the serum statistically significant? What is the value of p?

Line 285 – Please provide some real data, what is the percent in literature compared with the current study? Are there any differences with respect to the different L-asp, regimes, etc?

Comments on the Quality of English Language

Minor english language editing is necessary. There are a few misspellings and some syntax errors. 

Author Response

Dear Reviewer 2, all of your questions and suggestions have been addressed in detail in the attached document under the tab "Answers to Reviewer 2".

Kind regards,

Corresponding Author: Jasna Lenicek Krleza

Corresponding Author: Jasna Lenicek Krleza

Reviewer 3 Report

Comments and Suggestions for Authors

In the present study, the author investigated analytical and clinical significance of determining L-asparaginase activity in children with hematological malignancies in Croatian center. However, there are some issues that needs to be address.

1.     In line 33 and 34, According to the latest published data, which published data? The published data is not cited in the study.

2.      In line 88, In this manuscript should be “in this study”.

3.     In line 46, 1960.-ies should be 1960’s.

4.     What is the significance of this study? Is there any therapeutic application of this study? Should be mentioned in the introduction.

5.     In line 184, The ratio of boys to girls is approximately the same (0.9:1) and the number of girls is slightly higher (27:30), in which group the ratio of girls is slightly higher, the statement is unclear.

6.     In figure 2a and 2b include the statistical value for comparison and the quality of graph should be improve like include standard deviation in all the bar graph, it would be better if use prism software to generate the graph.  

7.     Statistical analysis in figure 3 is missing.

Comments on the Quality of English Language

 Extensive editing of English language required

Author Response

Dear Reviewer 3, I am attaching a tabular representation containing detailed answers to your questions and suggestions. Thank you for your valuable feedback.

Kind regards,

Corresponding Author: Jasna Lenicek Krleza

Round 2

Reviewer 1 Report

Comments and Suggestions for Authors

I accept all but one reply and modifications of the authors. However, the statement in Materials and Methods, that: "The result comparability was tested by using MedCalc® Statistical Software (MedCalc Software Ltd, version 22.017, Ostend, Belgium; https://www.medcalc.org; 2024), which was used for statistical tests of differences and/or comparison between the examined groups, as well as for graphs with all analytical and statistical data. The statistical significance of the difference is defined at the level of probability P< 0.05." is obviously true but not sufficient. MedCalc® is a statistical software that performs certain forms of statistical analysis. Yet, the type of statistical tests which were used (e.g. t-test, paired t-test, some of the non-parametrical tests, etc.) should be described.

Author Response

Dear Reviewer 1, Thank you for your many suggestions and we accept your last (Round 2)
comment and suggestion.
Changes have been made to the statistical and
analytical methods in the
Materials and methods section, point 2.4.
(added text, highlighted in yellow, lines 176-188): "....which was also used to test the differences between the study groups
using a t-test. Thus, the
t-test for the mean and median was performed to
test the difference between
the age groups compared to diagnosis (Table 3)
as well as to test the significance of the difference between the mean values
of L-
asp activity in the TDM points (Table 4). The comparison test of
proportions was used to test the significance of the difference in the
occurrence of toxic effects of L -asp therapy depending on the type
of preparation (Table 3). Summary statistics (descriptive data) were
calculated using the same MedCalc software program and are presented
in Tables 4 and 5. G
raphs (Figures 1 and 2) were also created using
the MedCalc program (
multi-variable graph, Box-and-Whisker plot with all data). Microsoft Excel 2016 was used to record all the patients data and
the
results obtained, as well as their calculation, analyzes and
also for
error bar graph (bar graph with the standard deviation marker
in Figure 3)."

Round 3

Reviewer 1 Report

Comments and Suggestions for Authors

Now only one point remained unanswered. Did you and if you did, how check for the normal distribution of values in groups compared with t-test.

Author Response

Dear Reviewer 1,

Thanks for the suggestion and we accept your last (Round 3) comment. In the Materials and methods section, point 2.4. Changes were made to the statistical and analytical methods (added sentence, highlighted in green, lines 185-186):

„The normality of the distribution of the individual variables was tested as part of the summary statistics using the Kolmogorov-Smirnov test.“

Kind regards

Round 4

Reviewer 1 Report

Comments and Suggestions for Authors

Modification and reply accepted.